# Cardiac Development, Cellular Composition and Function: From Regulatory Mechanisms to Applications

**DOI:** 10.3390/cells14171390

**Published:** 2025-09-05

**Authors:** Huan-Yu Zhao, Jie-Bing Jiang, Shu-Na Wang, Chao-Yu Miao

**Affiliations:** Department of Pharmacology, Second Military Medical University/Naval Medical University, Shanghai 200433, China; zhaohuanyu1128@smmu.edu.cn (H.-Y.Z.); jbjiang@smmu.edu.cn (J.-B.J.); snwang@smmu.edu.cn (S.-N.W.)

**Keywords:** cardiogenesis, myocardial cell function, cell therapy, gene therapy, cardiac organoid, tissue engineering, regenerative medicine

## Abstract

Cardiogenesis and heart cell composition and function constitute fundamental areas of cardiovascular medicine research, and exploring their underlying mechanisms is closely tied to the goals of precision medicine. This review comprehensively examines the composition and functions of the heart from embryonic organogenesis to maturity, and highlights the main breakthroughs of treatment strategies associated with these processes. By elaborating on the spatiotemporally specific signaling pathways and transcriptional networks that drive heart organogenesis and progenitor cell fate determination during the pivotal stages of cardiac development, and by systematically presenting the molecular biomarkers and functional characteristics of the principal cell types in mature heart, the latest advancements in related applications are summarized, with a particular emphasis on breakthroughs in gene/cell therapy, organoid development, and tissue engineering and regenerative medicine. This paper provides a theoretical foundation for precision interventions and regenerative medicine in cardiovascular disease using an axis that integrates cardiogenesis, cellular architecture, and therapeutic translation.

## 1. Introduction

Heart organogenesis is a highly complex and precisely orchestrated process encompassing the orderly development of several critical stages, such as primitive heart formation, shape establishment, internal compartmentalization, and conduction system development [1,2]. These developmental events are rigorously regulated by spatiotemporally coordinated gene expression networks, signaling pathways, and the interactions among various cell types, which collectively establish the foundation for a functional, mature heart. Dysregulation at any developmental node may disrupt normal cardiogenesis, potentially leading to congenital heart diseases (CHDs) [3]. After birth, the fetal heart transitions from maternal-placental dependence to autonomous cardiovascular function following cord clamping and the first breath. Thereafter, the myocardium enters a prolonged maturation phase characterized by cellular composition refinement and functional specialization to meet postnatal metabolic demands. Various types of cellular subsystems, such as cardiomyocytes, fibroblasts, and endothelial cells, demonstrate functional integration through paracrine communication and mechanical coupling, collectively sustaining the normal function of the heart [4,5]. Abnormal heart development and mature heart dysfunction can lead to a wide range of cardiovascular diseases, threatening human health. A thorough comprehension of the developmental processes and structural features of both embryonic and mature hearts is essential for elucidating heart disease pathogenesis and advancing cardiovascular disease treatments. Emerging precision medicine approaches, including cell therapy [6,7], gene therapy [8,9,10], organoid modeling [11,12,13], tissue engineering [14], and regenerative medicine [15], have been developed. However, current research lacks a systematic synthesis of development progress, mature characteristics, and related therapeutic applications.

This review aims to provide an in-depth delineation of current research on embryonic cardiogenesis, the structure and function of the mature heart, as well as advanced treatment strategies and technologies for cardiac diseases based on key regulatory mechanisms, thereby offering inspirations and references into cardiac regenerative medicine, disease research, and clinical therapeutic applications.

## 2. Heart Organogenesis

The heart, as the first functional organ during human development, initiates at the gastrula. Specifically, by the end of the second week of embryonic development (Carnegie Stage [CS] 7), cells have to make major phenotypic decisions in a spatially and temporally dependent manner to form three germ layers (ectoderm, mesoderm, and endoderm), from which specific body systems will subsequently develop [16]. The heart is primarily specified differentiation derived from the bilateral sheets of cells in the lateral part of mesoderm called lateral plate mesoderm [17,18]. The lateral plate mesoderm can also be further divided into two parts, splanchnic mesoderm and somatic mesoderm, in which the splanchnic mesoderm forms the cardiogenic heart crescent and begins the story of heart development [19]. The embryonic heart development can be roughly divided into four stages: (1) primitive heart formation; (2) shape establishment; (3) internal compartmentalization; and (4) conduction system development. The correct process of cardiac development is the basis for the correct cellular composition, structure, and function of the heart in the subsequent stage.

### 2.1. Primitive Heart Formation

At the third week of development (CS 8), mesoderm-derived cardiac progenitor cells give rise to a cardiac crescent and are broadly classified into the first heart field (FHF) and the second heart field (SHF) based on the predominant expression of multiple transcription factors, although no single marker can unequivocally distinguish between them (Figure 1A) [20,21]. Cells within the crescent differentiate into cardiomyocytes or endocardial cells [22,23]. Cardiomyocyte differentiation is promoted by endoderm-derived signals, including bone morphogenetic protein (BMP), fibroblast growth factor (FGF), and WNT signaling [20,21,24], while endocardial specification is initiated by the transcription factor *NKX2-5* (Figure 1A) [23,25]. Sox17 also promotes an endocardial bias in progenitors (Figure 1A) [26]. These signaling pathways and transcription factors function with spatiotemporal precision. BMP and WNT signaling further support primitive heart tube formation [27], chamber septation [28], and cardiomyocyte proliferation and differentiation [29,30]. *NKX2-5* also regulates hematopoietic precursors via Notch/RA signaling to generate macrophages essential for valve development [31]. Mutations in *NKX2-5* are associated with congenital heart defects, arrhythmias, cardiomyopathy, and hypothyroidism [32,33]. Therefore, precise regulation of genes and signaling pathways is essential for early cardiac development; even minor errors can cause structural and functional abnormalities. Whereafter, embryonic folding remodels the cardiac crescent into two endocardial tubes into a single primitive heart tube at the end of the third week (CS 9, Figure 1B) [2,25]. This structure comprises an endocardial layer and two or three cardiomyocyte layers, separated by cardiac jelly (Figure 1C). SHF maintains continuity with the FHF during development and adds cardiomyocytes to the developing heart [25]. Anomalies during this stage can lead to congenital heart diseases, such as atrial or ventricular septal defects.

### 2.2. Shape Establishment

From the fourth week (CS 10), the heart tube begins to swell, twist, and turn rightward to develop the atria, ventricles, and outflow tract (OFT) (Figure 1C,D) [1,2]. Cardiac jelly is replaced by immature trabeculae formed via endocardial protrusions (Figure 1C). Trabecular compaction, completed by CS 22 (Figure 1E), involves coordinated cardiomyocyte proliferation, migration, contractile function, metabolism, and interactions with ECs, fibroblasts, pericytes, Purkinje fibers, and other lineages, influencing cardiac structure and function upon maturation [34,35]. Activation of Notch signaling alleviates mechanical tension in compact layer cardiomyocytes and reinforces cell fate [35,36], while NRG1 and Notch1 synergistically promote extracellular matrix (ECM) remodeling, promoting synthesis and degradation, respectively [36,37,38]. Additional pathways, including BMP, FGF, and TGF-*β*, along with transcription factors such as *NKX2-5*, *GATA4*, and *Isl1*, further modulate trabeculation and compaction. The Hippo-YAP/TAZ pathway also plays a key role in embryonic cardiomyocyte proliferation and differentiation. Inhibition of the Hippo pathway, such as through deletion of adaptor proteins Salvador 1 (SAV1) or large tumor suppressor 1/2 (Lats1/2), induces beyond normal cardiac size and ventricular septal defect [39], whereas loss of Yes-associated protein (YAP) leads to impaired cardiomyocyte proliferation, cardiac hypoplasia [40].

From CS 10, the pro-epicardium contacts and envelops the heart tube, forming the epicardium. A subset of epicardial cells undergoes epithelial–mesenchymal transition (EMT) to form a mesenchymal layer, where coronary arteries develop (Figure 1C) [41,42]. Studies have demonstrated that WT1 regulates this process through modulation of the WNT/β-catenin signaling [43]. WT1 directly interacts with β-catenin, thereby regulating the expression of downstream target genes and influencing the fate determination of epicardial cells. WT1 also modulates retinoic acid (RA) signaling, a key morphogen in heart development that is essential for cardiac chamber septation and coronary artery formation [44]. Additionally, WT1 and the Hippo-YAP/TAZ pathway cooperatively regulate, ensuring that an appropriate number of cells contribute to the formation of the coronary artery tree and the development of the myocardial layer. Coronary smooth muscle cells and vascular adventitial fibroblasts originate from the proepicardium, while the coronary endothelium arises from the ventricular endocardium, septum transversum/proepicardium, and sinus venosus [45]. Studies have shown that genetic deletion of the Hippo pathway downstream, Lats1/2 or YAP/TAZ, and WT1 leads to abnormal coronary artery and myocardium development [39,46]. As the coronary tree completes development, the two coronary arteries form through angiogenesis and tap into the aortic lumen; abnormal OFT development could lead to erroneous connections [47]. Some mesenchymal layer cells invade the myocardium to become fibroblasts, constituting the fibrous skeleton of the heart and providing the necessary signals for cardiomyocytes development (Figure 1D) [25]. Some cardiac fibroblasts in the interventricular septum and RV originate from endocardium through endothelial-to-mesenchymal transition (EndMT) [48].

Cardiac progenitor specialization is spatiotemporally regulated: FHF cells typically form the primitive left ventricle (LV), while SHF cells contribute to the primitive right ventricle (RV), atrium, and OFT (Figure 2A) [49]. LV progenitors express *Tbx5*, *Nkx2-5*, and *Gata4*, jointly inducing cardiac genes such as *Hand1*, *Gja5*, *Mef2c*, and *Mlc2v* expression (Figure 2A) [50]. The anterior part of SHF, expressing Isl1 and Gata4, develops into the RV and activates Mef2c, leading to genes such as *Hand2*, *Hey2*, and *IRX4* expression (Figure 2A) [51]. RA signaling via retinoic acid receptor (RAR) and *COUP-TF II* promotes atrial identity and increases the expression of atrial CMs genes such as *Myl4*, *Sln*, *Gja5,* and *Nppa* (Figure 2A) [52,53]. COUP TF II regulates *ID2* for OFT development and suppresses *Hey2* and *Irx4* (Figure 2A) [54]. Despite symmetric atrial development, the left-sided identity is determined by *Pitx2c* [55]. High doses of RA can cause atrial enlargement and ventricular contraction, leading to congenital heart diseases [53].

### 2.3. Internal Compartmentalization

The linear heart is divided into four chambers through the atrial septation, atrioventricular valve, and ventricular septation from embryonic weeks 4 to 8 (CS 10–22, Figure 1D,E) [1]. This is a complex process that facilitates the initial disordered blood flow to separate into the systemic and pulmonary streams. The septum is expanded and reinforced by the ECM deposition between the endocardium and myocardium, forming cushions in the AVC and OFT, subsequently overlayed by the mesenchymal cells derived from endocardial transformation [56,57]. These structures are overlaid by mesenchymal cells derived from endocardial transformation via Notch, BMP, and TGFβ signaling (Figure 1D) [58]. The absence of YAP/TAZ results in ventricular wall thinning and defective development of the endocardial cushion, ultimately contributing to congenital heart defects [59]. The primary atrial septum develops from atrial cell proliferation at CS 12, connecting to the AVC layers through the mesenchymal cap and dorsal mesenchymal protrusion. From the end of the sixth week (CS 17), the secondary atrial septum appears through the muscle wall of the right atrium, folded down into the atrium to cover the entire secondary atrial foramen. It continues until birth, with the primary atrial septum being pushed toward the secondary septum under the blood pressure difference between the left and right atria, thus further preventing blood flow from the right to the left side [1]. The septum separating the LV and RV is composed of both the myocardial ventricular septum and the membranous septum. The ventricular intermediate cells form the myocardial ventricular septum via the addition of cells and extending [60]. Membranous septum formation commences during the second half of the sixth week (CS 19), and is mainly derived from the ventricular foramen separated by atrioventricular cushions and the septal OFT ridge [1,25].

### 2.4. Conduction System Development

All myocardial cells can conduct cardiac impulses; however, a specialized group of cells generates these impulses. This group forms the cardiac conduction system, mainly including the sinoatrial node, atrioventricular node (AVN), atrioventricular bundle branches, and Purkinje fibers. It begins to develop with the formation of the primitive heart tube (CS 10). At this stage, the primitive heart tube contracts due to the automaticity of cardiomyocytes at the venous pole (Figure 2B) [61]. All components of the cardiac conduction system originate from common cardiac progenitor cells and are tightly regulated by genes such as *NKX2-5*. Studies show that after the *Nkx2-5*-positive heart tube formation, it gives rise to an *NKX2-5*-negative phenotype in the sinoatrial node and the sinus horns through the inhibition of pacemaker channel gene *HCN4* and *T-box* transcription factor gene *TBX3* [55,62]. *TBX18* controls sinoatrial node head formation from mesenchymal precursors, followed by *TBX3* activating the pacemaker gene program [63,64,65]. Gene regulatory mechanisms are crucial for establishing the correct cardiac pacemaker and defining the atrium–sinoatrial node boundary. Cellular signaling pathways coordinate and drive these changes. For example, WNT5b promotes pacemaker progenitor differentiation by activating *Isl1* and *Tbx18* while inhibiting *Nkx2-5* (Figure 2B) [66]. Conversely, the WNT/β-catenin pathway disrupts progenitor cell fate, leading to *NKX2-5^+^* atrial cardiomyocyte lineage differentiation. Notch1 signaling absence significantly reduces *HCN4*, *TBX3*, and *TBX5* expression, affecting sinoatrial node development (Figure 2B) [67]. The RHOA-Rock pathway restricts the pacemaker gene program to the right sinus venosus side [68].

The electrical impulse travels through the atrial internodal pathways to the AVN, located at the right posterior atrial septum, where it undergoes a brief delay. The AVN primordium forms from endocardial cushion cells undergoing EndMT, followed by mesenchymal cell proliferation to populate the AV cushion and generate the AVN [69,70]. This process is orchestrated by signaling pathways and cardiac transcription factors, which activate the nodal gene program while suppressing the working myocardial gene program [71]. Like sinoatrial node development, *NKX2-5* and *TBX* family members play crucial roles. BMP2 activates *TBX3* and *TBX2* in the AVC to promote AVN formation, valve development, and atrioventricular insulation, while maintaining slow conduction (Figure 2B) [72,73,74]. Notch signaling regulates AVN functional maturation (Figure 2B); its absence results in loss of slow-conducting cell specificity and atrioventricular conduction delay [75].

The electrical impulse travels down through the bundle of His, located at the top of the interventricular septum. The His bundle progressively differentiates from the AVN, extends downward, and branches into the left and right bundle branches, which connect to Purkinje fibers to initiate ventricular contraction (Figure 2B). The ventricular conduction system ensures coordinated contraction via rapid depolarization mediated by sodium channels (Nav1.5) and gap junctions (CX40, CX43). *TBX5*, along with *NKX2-5*, *TBX3*, and *TBX2*, plays a key role in ventricular conduction system development (Figure 2B). *TBX5* absence impairs rapid conduction and causes arrhythmia due to the reduced expression of Scn5a and Gja5, the key mediators of the rapid conduction gene program (Figure 2B) [76,77]. *TBX5*, along with *NKX2-5* and *ID2,* form a transcriptional network driving ventricular cardiomyocyte differentiation into specialized conduction cells (Figure 2B) [78]. Similarly, *IRX3* interacts with *TBX5* and *NKX2-5* to regulate VCS-enriched genes such as *Gja* and induces the fast conduction gene program (Figure 2B) [79]. Notch activation in the ventricles converts cardiac progenitors into specialized conduction-like cells [80] by triggering juxtacrine signaling between fibroblasts and cardiomyocytes, promoting ventricular trabecular development via downstream regulators such as BMP10 and NRG1 (Figure 2B) [81,82].

## 3. Cell Composition and Function

The developmental process of the heart establishes the foundation for the function of the mature heart. 75–80% of human fetal (8–12 weeks of embryonic development) cardiac cells are cardiomyocytes, while non-myocytes make up only 20–25% [59]. The embryonic heart progressively establishes the cellular composition and functional network of the mature heart through precise lineage differentiation and dynamic interactions with the microenvironment. Monika et al. identified 11 major types and distributions of cardiac cells using single-cell and single-nucleus RNA sequencing combined with multiplex single-molecule fluorescence in situ hybridization (smFISH) in half a million single cells from 14 adult cardiac tissues (Figure 3A) [83].

### 3.1. Cardiomyocytes

Ventricular and atrial cardiomyocytes, derived from the SHF and FHF, respectively, are the fundamental units of the myocardium and drive cardiac contraction. The proportions of cardiomyocytes in atrial and ventricular regions of the mature heart are 30.1% and 49.2% [83]. Cardiomyocytes primarily proliferate during the embryonic period, whereas they lose this ability after human birth, growing instead through physiological hypertrophy (Figure 3B) [84]. Hippo inhibition or the downstream YAP activation enhances cardiomyocytes’ survival, proliferation, and regeneration to protect the heart [39,85,86]. The mature ventricular cardiomyocytes (vCMs) predominantly express myosin heavy chain β (βMHC, encoded by *MYH7*), which drives structural and organizational changes in the myofibrils, along with MLC2V (encoded by *MYL2*), a marker specific to vCMs. In contrast, the mature atrial cardiomyocytes (aCMs) are characterized by the expression of MLC2A (encoded by *MYL7*) (Figure 3B) [49,87]. The maturation encompasses various intricate transitions and precision controls. The maturation of sarcomere organization is marked by the transition from slow skeletal muscle troponin I (ssTnI) to cardiac troponin I (cTnI), along with the splice isoform shift of Titin [88]. In fetal myocardium metabolism, reliance on the glycolytic pathway is evident, where anaerobic metabolism-related genes such as hexokinase, glucose transporter-1 (GLUT-1), and lactate dehydrogenase A (LDHA) are predominantly regulated by hypoxia-inducible factor-1α (HIF-1α) [89,90,91]. Postnatally, as respiration initiates and lung expansion occurs, the metabolic pattern transitions from glycolysis to oxidative phosphorylation (fatty acid β-oxidation) [90,91,92]. During this metabolic shift, the reduction in HIF-1α results in decreased expression of hexokinase, GLUT-1, and LDHA; in contrast, the PPAR family is activated, along with its downstream signaling pathways, to fulfill the increased energy demands (Figure 3B). Thus, metabolic shift is thought to be one of the inducers in myocardial maturation. Cardiomyocyte metabolic switch and end of proliferation largely depend on the timing of the species. Mammals with limited regenerative capacity, such as humans and mice, exhibit a short and tightly regulated metabolic transition window, making them widely used models for investigating the mechanisms of cardiomyocyte proliferation exit. In contrast, species with robust regenerative capabilities, such as zebrafish, may maintain prolonged metabolic “plasticity” in their cardiomyocytes, offering valuable insights into the identification of pathways that promote cardiac regeneration [93]. Additionally, cardiac maturation results in significant changes in electrophysiological properties, transitioning cardiomyocytes from spontaneous contraction to reliance on the cardiac conduction system and pacemaker cell activity. The strength and rhythm of contraction in mature cardiomyocytes are precisely controlled by the electrical coupling of Ca^2+^ ion channels with action potentials. Voltage-gated sodium channels mediate Na^+^ influx and membrane depolarization, which trigger action potentials and subsequently activate L-type calcium channels (Cav1.2), leading to stable Ca^2+^ action currents [94]. The resting membrane potential of mature cardiomyocytes is approximately −90 mV, which is lower than that observed in immature cells, and the disparity might be attributed to the genes *KCNJ2* and *KCNJ12*, which encode the potassium channel subunits Kir2.1 and Kir2.2 (Figure 3B) [4,94,95]. Desmosomes and adherens junctions coordinate cardiomyocyte contraction and electrical propagation by connecting adjacent cells mechanically and transmitting electrical signals and small molecules [96].

### 3.2. Fibroblasts

The proportion of fibroblasts in the mature atria and ventricles is 24.3% and 15.5%, respectively [83], and increases continuously after injury and during aging [97]. Their main sources are pre-epicardium differentiation and EMT/EndMT of the mesenchymal layer or endocardium during embryonic development (Figure 4A) [25,48]. Transcription factor 21 (*Tcf21*), Wilms’ tumor gene 1 (*WT1*), T-box transcription factor 18 (*Tbx18*), and platelet-derived growth factor receptor alpha (PDGFR-α) are markers of embryonic fibroblasts, whereas only *Tcf21* and PDGFR-α are expressed by mature fibroblasts [5,98,99,100,101]. It is essential for maintaining cardiac structure and mechanics, coordinating collagen network production and remodeling for conductivity and rhythmicity (Figure 4A) [102]. Fibroblasts also maintain ECM homeostasis through autocrine and paracrine mechanisms. They secrete matrix metalloproteinases (MMPs), which disrupt the ECM network, enhance inflammatory cell infiltration, and promote cytokine production, thus stimulating fibroblast migration and differentiation, leading to the deposition of new ECM [97,103]. Transduction signals of myocyte–fibroblast crosstalk are also important for fibroblast proliferation, cardiomyocyte growth, and ECM turnover [103]. Fibroblasts may serve as barriers, potentially slowing down or interrupting cardiac electrical excitability; they also function as mechanical sensors and electrical signal regulators, actively participating in the electrophysiological regulation of cardiomyocytes through mechano-electrical feedback [104,105]. This feedback may occur through direct electrical coupling with cardiomyocytes via gap junctions and/or tunneling nanotubes, thereby actively modulating the electrophysiological properties of cardiomyocytes [106,107]. Although this process facilitates electrical synchronization under physiological conditions, it may predispose to arrhythmogenesis under pathological circumstances. On the other side, fibroblasts act as sentinels, rapidly responding to myocardial injury by initiating appropriate inflammatory and repairing reactions [108,109]. They then extensively proliferate and transform myofibroblasts, which actively secrete matrix proteins and contribute to scar formation [110]. The cell transforms from resident fibroblasts into active fibroblasts, myofibroblasts, and matrifibrocytes during the process described above [111]. Specifically, upon injury, quiescent fibroblasts are transitioned into an activated state, which begins to proliferate extensively and migrate to the injury site, acting as the initial responders in the tissue repair process [109]. Myofibroblasts differentiate from activated fibroblasts through multiple signaling pathways and serve as key effector cells in cardiac fibrosis. They express α-smooth muscle actin, which integrates into stress fibers to generate strong contraction, aiding scar formation and preventing cardiac rupture after tissue necrosis [112]. Additionally, hematopoietic progenitor cell recruitment and EC transformation are also thought to be important sources of myofibroblasts [113,114]. Activation of the Hippo-YAP/TAZ pathway enhances fibroblast differentiation into myofibroblasts and stimulates the secretion of pro-fibrotic and inflammatory mediators, such as IL-33, CCL2, and CCN3 [115,116]. However, prolonged activation of myofibroblasts leads to pathological remodeling and declining cardiac function via excessive secretion of ECM. Matrifibrocytes, a relatively newly identified fibroblast-like cell state, play a critical role in sustaining scar integrity for cardiac function preservation [117].

### 3.3. Endothelial Cells (ECs)

Mature cardiac ECs are roughly divided into three types—endocardial ECs, microvascular ECs, and coronary ECs—according to their location. Endocardial ECs, derived from mesodermal cardiogenic precursors, act as physical barriers within the cardiac chamber and connect to the vasculature through de novo vasculogenesis (Figure 4B) [23]. They also participate in cardiomyocyte transformation and development by secreting signals that induce trans-differentiation into Purkinje fiber cells or transmit essential signals for proper trabecular myocardium formation (Figure 4B) [61,118]. Moreover, specific regions of endocardial ECs undergo EndMT to form endocardial cushions (Figure 1D and Figure 3B) [119,120]. Microvascular and coronary ECs originate from endocardial progenitor cells and endocardial cells to form a capillary network in the myocardium (Figure 4B) [121]. They serve as structural barriers between blood and vessel walls, regulate vascular permeability and tension, supply oxygen and nutrients to cardiomyocytes, and communicate with other cell types through autocrine and paracrine signaling. Moreover, ECs modulate the growth and contraction of cardiomyocytes via paracrine signaling and secretion of various factors. For example, nitric oxide (NO) secreted by ECs alleviates cardiac hypertrophy through the NO-sGC-cGMP pathway [122], while endothelin-1 (ET-1) and neuregulin-1 (NRG-1) secretion promote hypertrophy [123,124]. Furthermore, ECs influence cardiomyocyte death by regulating apoptosis [125]; conversely, cardiomyocytes can stimulate endothelial neovascularization after ischemia [126]. On the other hand, the secretion of TGFβ1, NO, vascular cellular adhesion molecule-1 (VCAM-1), and intercellular adhesion molecule 1 (ICAM-1) by ECs stimulates fibroblast proliferation and migration, transforming fibroblasts into myofibroblasts, which allow more extracellular matrix secretion to promote myocardial fibrosis [127,128]. Hippo-YAP signaling in ECs functions differently depending on the context. YAP activation usually promotes proliferation, inflammation, and atherosclerosis, yet its absence in ECs increases lung inflammation and mortality in sepsis models, suggesting that YAP activity must be tightly regulated [129,130]. Its upstream kinases, such as MST1, regulate vascular function and disease progression through YAP-independent pathways, including connexin 43 [39].

### 3.4. Smooth Muscle Cells (SMCs)

SMCs are fundamental components of the tunica media in coronary and arteriolar walls (Figure 5A). The main function is to keep vascular contractility, adjusting vascular lumen diameter and pressure, maintaining vascular tone and integrity, and distributing blood volume to other organs [131,132,133]. SMCs of cardiac coronary arteries are mainly differentiated from embryonic epicardial progenitor cells between CS 16–18 [25,134]. Arterial vascular SMCs (VSMCs) contract similarly to cardiomyocytes, albeit at a slower rate, and play a crucial role in regulating blood flow [131,135]. VSMCs exhibit high phenotypic plasticity, primarily characterized by a contractile phenotype under physiological conditions (Figure 5A). However, high stretch stimulation may trigger intracellular molecular changes in VSMCs, transforming them into a secretory phenotype through dedifferentiation (Figure 5A) [136,137,138,139]. The transformation enhances the abilities of proliferation, migration, and secretion, while weakening contraction. Many factors (such as Notch and downstream signals) maintain VSMC stability [140,141]. Additionally, platelet-derived growth factor BB (PDGF-BB) is a key regulator in promoting VSMC dedifferentiation via activating multiple pathways and reducing the expression of contractile markers (Figure 5A) [142,143]. Inversely, overexpression of *miR-214* prevents PDGF-BB-induced transformation [144]. Various inflammatory factors also lend a hand to the transformation of secretory VSMCs, and synthesize massive amounts of ECM [145,146]; in turn, the changes of the ECM microenvironment could alter VSMCs [137,147,148,149]. Thus, the differentiation/dedifferentiation of VSMCs is a multifactorial process, during which they can take on the characteristics of other cell types, such as myofibroblasts, osteoblasts, and macrophages [150,151,152].

### 3.5. Pericytes

Pericytes, another type of mesenchymal cell, are embedded into the basement membrane of ECs in arterioles, venules, and capillaries that monitor and stabilize EC maturation through physical contact and paracrine signals for regulating vascularization (Figure 5B). Research has demonstrated that the proportion of mural cells, including pericytes and SMCs, varies slightly between atrial and ventricular tissue, with values of 17.1% and 21.2%, respectively [83]. Pericytes maintain the endothelial barrier via secreting multiple cytokines, such as transforming growth factor-beta (TGF-β), vascular endothelial growth factor (VEGF), sphingosine 1-phosphate (S1P), angiotensin (Ang-1 and Ang-2), to enhance adhesion and tight junction [153]. Angiopoietin-1/2 secreted by ECs promotes pericyte dissociation from vessels, aiding their migration. This migration relies on ECM modification by MMPs and proteases. Pericyte-derived VEGFA then stimulates endothelial proliferation and survival, leading to sprouting and neovascularization; however, TGFβ secretion inhibits further proliferation, while endothelium-derived PDGF-BB and HB-EGF recall pericytes to stabilize new vessels (Figure 5B) [154]. The pericyte coverage of cardiac ECs ranges approximately from 1:2 to 1:3, a ratio significantly lower than that observed in other organs. This discrepancy may reflect physiological demands, as microvessels must support frequent, efficient material exchange to maintain the continuous contraction of myocardial cells. Although pericytes are known for maintaining vascular homeostasis via controlling vascular tone and stabilization, regulating permeability and angiogenesis, they have been far less studied than other cardiac cells (Figure 5B) [155,156,157]. The origin of pericytes remains debated. Some suggest they originate from mesenchymoangioblasts and are considered as vascular stem cells [158,159,160], while others propose they share a similar phenotype with MSCs and co-originate from endocardial progenitor cells [161]. Currently, no specific marker exclusively identifies pericytes; instead, a combination of markers such as NG2, PDGFRβ, and α-SMA is typically used [154].

## 4. Breakthroughs in Treatment Strategies

The critical transcription factors, spatiotemporal signaling pathways, and epigenetic modifications governing cardiac development and maturation mentioned above form a robust foundation for the intervention and treatment against cardiovascular diseases (Figure 6). Building on this knowledge, significant advancements have been achieved in the development of treatment strategies for heart diseases.

### 4.1. Cell Therapy

The cellular composition of developing and mature hearts plays a pivotal role in cardiac cell therapy. Understanding this composition provides insights for selecting cell types and designing therapeutic strategies. Progenitor and mature cells from various origins, including non-cardiac (e.g., skeletal myoblasts [162], mesenchymal stem cells [163]), cardiac (e.g., c-Kit^+^ cells [6], cardiosphere-derived cells [164], epicardium-derived progenitors [165]), and pluripotent stem cells (e.g., ECSs, iPSCs [7]), have been transplanted into damaged cardiac tissue. Each cell type has unique advantages. Early heart disease therapies using skeletal muscle myoblasts improved ejection fraction in animal models but were discontinued due to arrhythmia risks caused by electrical mismatches and incomplete differentiation into cardiomyocytes [162,166]. This observation highlights the critical importance of cellular “cardiac adaptability”. Moreover, early clinical trials of bone marrow-derived mononuclear cells (e.g., CHART-1, FOCUS-CCTRN, MiHeart, and TAC-HFT) and adipose-derived regenerative cells (e.g., ATHENA, Danish phase II, and SCIENCE) in heart failure patients did not demonstrate significant improvements in left ventricular function, although certain studies reported beneficial effects on life quality or a decrease in the incidence of sudden cardiac death [167]. Hence, these findings encouraged further research into more effective cell types and optimized preparation protocols. Mesenchymal stem cells (MSCs), extractable from bone marrow, adipose tissue, and other organs, are suitable for allogeneic transplantation due to low MHC II expression [168]. In recent years, several well-designed clinical trials of MSCs, including ixCELL-DCM [169], MSC-HF [170], CONCERT-HF [171], DREAM-HF [172], and SENECA [173] have demonstrated that while cell therapy does not consistently enhance left ventricular ejection fraction, it can significantly reduce the incidence of heart failure-related clinical events, improve quality of life, and exhibit long-term safety. Safety control has consistently represented a significant challenge in current cell therapy, and thus, these clinical trials may serve as valuable references for future advancements in technical methodologies. Notably, the RIMECARD trial was the first to demonstrate that intravenous infusion of umbilical cord-derived MSCs can lead to sustained improvements in left ventricular ejection fraction and overall cardiac function [174]. Co-delivering bone marrow-derived MSCs and c-Kit^+^ cells enhances right ventricular function in rats with pressure overload and improves ischemic heart failure in patients [6,171]. Additionally, c-Kit^+^ cell transplantation contributes to vascular remodeling and neointima formation [175]. Cardiospheres-derived cells, containing c-Kit^+^ cells, connexin 43-positive intermediate layers, and CD105-expressing outer layer, show superior myocardial repair in rodent models compared to single stem cell therapies and have advanced to clinical trials [176,177,178]. Therapy strategies based on WT1 primarily employ WT1^+^ progenitor cells derived from the epicardium for cardiac repair, relying on their multipotent differentiation potential to generate cardiomyocytes, smooth muscle cells, and endothelial cells [179]. Epicardium-derived progenitor cells emerge during the EMT in CS 10 [180], gradually losing proliferative capacity to become dormant [181]. Upon activation, they release factors, such as TGFβ, PDGF, FGF, IGF, BMP, RA, Notch, and extracellular matrix proteins to directly promote cardiac tissue regeneration [165,182,183]. Experiments showed that pretreatment with thymosin β4 (Tβ4) can activate quiescent progenitor cells and enhance angiogenesis [184,185]. Collagen patches containing Fstl-1, a protein from these progenitor cells, partially improve myocardial contractile function [186]. Although these cardiac progenitor cells are theoretically capable of differentiating into specific cell types, their multipotency and functional contributions remain subjects of debate. Numerous studies have failed to reproduce findings of their differentiation into functional cells, suggesting that their therapeutic effects may be mediated primarily through paracrine mechanisms. The precise regulation of their differentiation fate, thus, might remain an unresolved challenge.

PSCs, including ESCs and iPSCs, represent the authentic source of “regenerative” potential, as they are capable of producing a substantial quantity of functional cells. ESCs can theoretically differentiate into all mesoderm-derived cell types for cardiac function. For example, mouse ESCs transplanted into infarcted sheep myocardium differentiated into cardiomyocytes and improved LV function [187], while human ESC-derived cardiomyocytes showed similar benefits in rat models [188,189]. The first relevant clinical trial, ESCORT [190], confirmed the safety of hESC-derived cardiovascular progenitors epicardial delivery in patients with severe ischemic left ventricular dysfunction and triggered the subsequent SECRET-HF trial of exosome research [191]. However, ESCs’ practical application is still limited by tumorigenicity, immune rejection, and ethical issues [192,193]. iPSCs offer advantages for patient-specific treatment. Studies indicate that implanting iPSC-derived cardiovascular progenitor cells or cardiomyocytes in rodent MI models improves cardiac function and reduces fibrosis [194,195]. Early clinical trials have reported the safety of human iPSC-derived cardiomyocytes in treating severe ischemic cardiomyopathy [190,196]. Safety represents a critical parameter in the design and evaluation of clinical trials. Moreover, the HECTOR, LAPIS, and the hiPSC-CM slice and BioVAT-HF trials have each investigated delivery methods such as transcatheter administration, intraoperative injection, and tissue patch implantation [167]. These clinical trials have not only preliminarily demonstrated the safety of cell therapy derived from iPSCs, but also indicated its potential to improve cardiac function. Although iPSCs have addressed the challenge of individualized therapy, safety concerns such as epigenetic memory and potential off-target effects associated with gene editing remain to be thoroughly evaluated over the long term. Therefore, the current research focus on iPSC cell therapy is likely transitioning from “whether it is safe” to “how to achieve more effective therapeutic outcomes”. Additionally, self-organizing cardiac organoids constructed by ESCs/iPSCs represent a promising new approach (see details below).

### 4.2. Gene Therapy

Gene therapy offers significant advantages over traditional drug therapy, such as modifying intracellular signaling pathways and gene expression [10]. Advances in gene therapy in cardiovascular disease reveal potential for treating cardiac developmental anomalies, congenital heart defects, myocardial infarction recovery, and heart failure mitigation [8,9]. Mutations in genes such as *TBX1*, *GATA4*, and *NKX2–5*, along with dysregulated signaling pathways during cardiomyogenesis, cause structural abnormalities in congenital heart diseases. Targeted gene therapy aimed at these regulators holds promise for correcting cardiac defects. For example, Tetralogy of Fallot, which involves abnormal expression of these genes, highlights how insights into cardiac development can guide therapeutic strategies [197]. CRISPR-Cas9-based gene editing shows promise in correcting mutations to restore cardiac function, with encouraging preclinical results [198]. After myocardial infarction, despite the heart initiating repair mechanisms, complete restoration of function remains challenging. Reactivating genes such as *VEGF* and *FGF*, crucial during cardiac development, can promote angiogenesis and tissue repair post-infarction [199,200]. Moreover, promoting cardiomyocyte proliferation while limiting excessive immune activation has consistently remained a central focus in myocardial infarction repair. Recent studies suggest that targeting WT1 or its downstream pathways offers the potential to achieve multiple therapeutic benefits simultaneously, including angiogenesis, immune regulation, and cell protection [201]. These effects are mediated through the direct regulation of pro-angiogenic factors such as VEGF and CD31, which promote angiogenesis and enhance blood perfusion; maintenance of the immunosuppressive functions of CD11b and Ly-6G, which limit excessive T-cell infiltration and inflammatory responses, thereby preventing immune-mediated cardiac damage; and reduction in apoptosis, which fosters a more conducive microenvironment for tissue repair [201]. Hippo-YAP/TAZ activation promotes heart regeneration [202,203]. Researchers found that the cardiac-specific expression of YAP-1SA or shRNA-mediated knockdown of *Sav1* (Hippo pathway adaptor) via the adeno-associated viral 9 (AAV9) vector effectively activates YAP/TAZ, thereby promoting cardiomyocyte proliferation and cardiac regeneration in mice [204,205]. It is worth considering whether sustained overactivation of Hippo-YAP signaling may lead to adverse effects, including cardiac excessive growth, tumorigenesis, or other complications. Therefore, more research and validation are still needed to fully evaluate their long-term safety. Conversely, there may also be instances of inadequate gene regulation. Thus, achieving precise and controllable gene expression remains an important area warranting further investigation. Furthermore, local subendocardial injection of AAV9-*Sav*-shRNA via catheter into the border zone of myocardial infarction in pigs could reduce scar formation and improve ejection fraction, without evidence of tumor formation or toxic reactions [206]. Based on these, regulating their expression or function through gene therapy may enhance cardiac repair and regeneration; however, the translational pathway from laboratory research to clinical application remains fraught with challenges (see below).

In therapeutic angiogenesis gene therapy, early studies primarily focused on ischemic heart disease, aiming to improve myocardial blood supply by promoting angiogenesis and/or remodeling existing vessels. For example, the ReGenHeart trial evaluates the efficacy and safety of Ad.VEGF^DΔNΔC^ gene transfer in refractory angina patients; the EXACT trial assesses the safety and preliminary efficacy of Ad.VEGF-All in similar patients; the AFFIRM trial evaluates the efficacy and long-term safety of Ad5.FGF-4 in severe angina patients, and the EPICCURE trial investigates the efficacy of modified RNA (*VEGF-A165*) in patients with moderate cardiac dysfunction [10]. Unlike angiogenesis gene therapy, which relies on secreted growth factors, heart failure treatment targets cardiomyocyte abnormalities. Trials targeting sarco/endoplasmic reticulum Ca^2+^ ATPase (SERCA2a), adenylyl cyclase VI, stromal cell-derived factor-1, and inhibitor-1 aim to enhance cardiac function through distinct molecular mechanisms [10]. Despite progress, challenges remain, including low gene transduction efficiency, immune responses to AAV vectors, off-target effects, and long-term expression stability [207]. The cardiac environment is highly heterogeneous and dynamically changing, particularly following heart failure or myocardial infarction. Targeting a single gene target, such as SERCA2a or FGF, as mentioned above, may be insufficient to address the complexity of the entire disease network, which could partially explain why certain early-phase clinical trials failed to achieve their primary endpoints. Although cardiovascular gene therapy currently lags behind other therapeutic fields and exhibits many of the aforementioned limitations, ongoing efforts and in-depth research are expected to enable gene therapy to play an increasingly significant role in the treatment of cardiovascular diseases.

### 4.3. Self-Organizing Cardiac Organoid Construction

Human self-organizing cardioid is a novel cardiac organoid model that recapitulates embryonic heart development and cell differentiation processes through signal regulation [11,12]. Since 2021, this field has seen rapid progress. The precise regulation of normal cardiac development provides crucial guidance for constructing these organoids. Current methods typically involve forming embryoid bodies from human pluripotent stem cells (hPSCs), either with or without biological scaffolds, followed by sequential differentiation into mesoderm, cardiac mesoderm, and cardiac lineages under various stimuli to replicate cardiac functions. Signals from other embryonic layers also play key roles in cardiac organoid development [13,208,209,210]. The method by Schmidt et al. involves separately forming distinct cardiac lineage organoids and integrating them artificially, resulting in more complex structures and mature functions [211]. Regardless of the method, precise signal regulation remains indispensable. For example, WNT and BMP signaling regulate cardiomyocyte differentiation, primitive heart tube formation, ventricular differentiation, and influence organoid size and morphology through different temporal and spatial sequences [11,12,212]. Abnormalities in transcription factors such as *NKX2-5*, *ET1*, *ISL1*, *TBX5,* and *FOXF1* lead to structural and functional defects in cardiac organoids [211,213,214], mirroring those observed during embryonic heart development. Advances in cardiac organoid technology not only guide the cultivation of self-organizing cardioids but also provide valuable insights into heart development research, helping to identify new cell types and refine our understanding of human heart development.

Self-organizing cardiac organoids, which closely mimic the natural heart, overcome in vitro research limitations by recapitulating complex cellular interactions and structural organization. The various existing construction models have been applied to study congenital heart defects [12], genetic disorders [213], cardiotoxicity [215], and injury-related conditions [11,211,214,216] (such as fibrosis and regeneration). They provide opportunities for developing cardiovascular disease models, elucidating disease mechanisms, evaluating drug safety and toxicity, and advancing cardiac regeneration and therapeutic transplantation. The significant potential of these cardiac organoids in detecting drug-induced cardiotoxicity, especially from chemotherapy drugs, such as temozolomide, doxorubicin, and ondansetron, has been validated [215,216,217,218]. Although no organoid approach has yet been able to achieve cardiac repair or regeneration in humans, scientists have not given up their research. Lee et al. demonstrated that heart organoids with microvascular networks can connect with host vasculature after subcutaneous transplantation in mice [219]. Similarly, epicardial organoids developed by Wang et al. integrated with adult mouse hearts, enabling organoid-derived cell migration into the myocardium [212]. Blood-generating heart-forming organoids produce hematopoietic progenitor cells with erythroid, myeloid, and lymphoid pluripotency, offering insights into cardiac hematopoiesis-based transplantation strategies [210].

Current self-organizing cardiac organoids remain immature and fail to fully replicate the structural complexity and functional maturity of the adult human heart. Key limitations include inadequate vascularization, irregular cardiomyocyte organization, and significant electrophysiological differences compared to mature tissue. The lack of standardized protocols and unified assessment frameworks further hinders reproducibility and comparability across studies. Integrating advanced technologies such as 3D bioprinting, engineered biomaterials, and co-culture systems offers promising pathways to enhance vascularization and functional maturation. Ultimately, interdisciplinary collaboration is essential for developing more physiologically relevant cardiac models for disease modeling and drug screening.

### 4.4. Tissue Engineering and Regenerative Medicine

In addition to self-organizing cardioids, the advancement of cardiac tissue engineering and regenerative medicine is also relevant to the understanding of the processes of heart development and maturation. The three-dimensional layered structure of the heart, including the myocardium, valves, and vascular network, self-organizes during development through intricate cellular interactions. Tissue engineering must leverage this developmental process to replicate the natural architecture via advanced techniques such as 3D bioprinting [220,221], special materials [222,223], and dynamic culture systems (e.g., fluid shear stress simulation) [224], using these approaches that mimic the cardiac physiological microenvironment for promoting cardiomyocyte proliferation and differentiation [225]. Meanwhile, the key signaling pathways (e.g., WNT/BMP), cell differentiation regulations, metabolic control, and microenvironmental regulatory mechanisms uncovered during heart development serve as direct guidance for the directed differentiation of stem cells and the design of bionic scaffolds in regenerative medicine [226].

Traditionally, bovine pericardial patches have been used for repairing cardiac defects. However, due to their biological inertness, they are susceptible to calcification, thrombosis, and inflammatory reactions, and cannot be degraded or absorbed by the human body. To overcome these limitations, scientists have developed novel reinforced cardiac patches for myocardial infarction and repair [227]. Exosomes exhibit strong tissue penetration, low immunogenicity, and high stability, while avoiding the tumorigenic risks and ethical concerns associated with traditional cell therapy, making them a highly promising therapeutic platform in cardiac regenerative medicine. The molecular composition of exosomes is highly complex and specific to their cell of origin, determining their functional diversity. Exosomes derived from different mesodermal progenitor cells carry distinct repertoires of bioactive molecules, leading to diverse functions in cardiac repair [228]. Notably, exosomes from cardiac progenitor cells are enriched in multiple *microRNAs*, such as *miR-210*, *miR-132*, *miR-214*, and *miR-146a-3p*, which exhibit key therapeutic effects, including the reduction in myocardial fibrosis and suppression of inflammation, as well as the enrichment of extracellular proteins involved in tissue repair pathways [228]. Mesenchymal stem cell-derived exosomes are enriched in anti-inflammatory and pro-angiogenic factors such as VEGF, FGF, and IL-10, which reverse ventricular remodeling and improve long-term cardiac function by modulating immune responses and promoting angiogenesis [229]. Engineered exosomes, such as those derived from KLF2-overexpressing endothelial cells, highly express therapeutic molecules such as *miR-486-5p* and can specifically target and regulate the PTEN-PI3K/Akt signaling pathway in cardiomyocytes to inhibit apoptosis [230]. This functional versatility allows researchers to select the most suitable exosome source for therapeutic intervention. However, it also carries potential risks. On the one hand, the complex composition of exosomes makes the key mechanisms unclear, leading to the inability to conduct clear quality control; on the other hand, exosomes from different sources and extracted by different methods vary greatly in composition and function, and the lack of standardized control processes makes industrial development difficult. From the perspective of multidisciplinary integration, exosomes demonstrate enhanced therapeutic efficacy and improved targeting precision when combined with tissue engineering strategies. For example, injectable pH-responsive conductive hydrogels can mimic the mechanical and electrophysiological properties of cardiac tissue and intelligently deliver exosomes to enhance cardiac repair after myocardial ischemia-reperfusion injury [231]; multiple microneedle patches have been engineered to deliver exosomes derived from multiple cell types, thereby attenuating inflammation, reducing infarct size, inhibiting fibrosis, and enhancing cardiac function [232,233,234]; and engineered exosomes carrying ischemic myocardium-targeting peptide significantly improve the targeting specificity of exosomes to the ischemic myocardium [235,236]. Furthermore, an advanced biomaterial platform, extracellular matrix–synthetic hydrogel hybrid scaffold, is capable of precisely mimicking the cardiac cell extracellular environment and independently modulating key extracellular matrix properties, such as ligand presentation and stiffness, to reverse the senescence state of cells [237]. As exosome-based cardiac therapy advances, personalized treatment is emerging as a key future direction, which involves reducing immune rejection risks using iPSC technology and improving exosome functions through gene editing, such as CRISPR/Cas9, to increase targeting precision and therapeutic effectiveness. At the same time, issues such as standardization and long-term safety need to be addressed.

On the other hand, the main challenges in cardiac regenerative medicine include effectively promoting cardiomyocyte proliferation, enhancing the survival and integration of transplanted cells in the injured myocardium, modulating the cardiac immune microenvironment, and delivering appropriate biophysical and biochemical signals to support tissue repair. Currently, interdisciplinary strategies that integrate cell therapy, biomaterials, molecular targeted therapy, and gene regulation remain among the most viable approaches to addressing these complex challenges. For example, fatty acid oxidation inhibition directly controls cardiomyocyte maturation and proliferation, with metabolism-induced partial reversal of maturation enhancing proliferation and activating cardioprotective mechanisms, including adaptive HIF1 signaling and αKG-dependent DNA damage repair [238]. However, further research is warranted to identify the inhibitors of fatty acid oxidation that do not cause liver damage, to promote myocardial proliferation [239]. Excitingly, recent studies have shown that transplanting healthy mitochondria into repair-capable macrophages, which can continuously release functional mitochondria after entering the infarcted area, enhances fatty acid metabolism and oxidative phosphorylation, thereby improving myocardial energy supply and restoring over 50% of cardiac contractile function following myocardial infarction [240]. In addition to serving as an energy source, the mitochondria promote the polarization of macrophages toward the anti-inflammatory M2 phenotype, enhance myocardial angiogenesis and extracellular matrix remodeling that reduced the myocardial infarct size and restored the ejection fraction to 76% of normal levels [240]. The integration of these fields has significantly propelled advancements in cardiac repair technologies. Looking ahead, it is essential to further elucidate the underlying developmental mechanisms, overcome existing technical challenges, and ultimately realize the regeneration and clinical translation of functional cardiac tissues. The technical challenges in tissue engineering, including immature cell functions and low transplantation integration rates, have driven researchers to revisit the latent regenerative potential within developmental programs, such as the proliferative characteristics of cardiomyocytes during embryogenesis. By employing strategies such as gene editing and dynamic mechanical simulation using biomaterials, it aims to reconstruct the developmental microenvironment, thereby hoping to overcome the limitations of adult heart regeneration.

## 5. Conclusions and Challenges

This article reviews the key processes of embryonic heart development, cellular composition, and functional characteristics of the mature heart. These are regulated by complex and precise mechanisms at cellular, genetic, and signaling levels. A deeper understanding of the signaling pathways and gene networks during cardiogenesis will help to clarify the mechanisms underlying diseases. Embryonic cardiogenesis helps elucidate the initial stages of heart disease onset; meanwhile, clarifying the main cellular composition and function of the mature heart provides potential cellular-level targets for the precise treatment of heart diseases. However, due to ethical constraints, our knowledge of the developmental processes and regulatory mechanisms underlying heart development remains fragmented and necessitates further in-depth exploration. Current treatments for heart diseases are continuously advancing based on existing mechanisms and scientific progress. Cell therapy leverages the differentiation potential of stem cells to potentially replenish damaged myocardial cells; gene therapy enables precise correction of defective genes and regulation of heart-related gene expression; organoid technologies simulate heart development and disease models, offering a robust platform for drug screening and mechanistic studies; and tissue engineering and regenerative medicine aim to construct functional heart tissues to replace damaged areas. Despite progress, challenges remain.

Except for PSCs, the underlying mechanisms of action in cell therapy remain largely unclear. It is increasingly evident that therapeutic effects may primarily stem from paracrine signaling through secreted cytokines, exosomes, and other bioactive molecules, promoting endogenous repair rather than true myocardial regeneration. This opinion reasonably accounts for the observation that while improvements in left ventricular ejection fraction are often limited, clinical symptoms tend to show significant amelioration. Although PSCs can differentiate multiple cells, their application is hindered by several critical challenges, including tumorigenicity, immune rejection (particularly with allogeneic ESCs), and high costs coupled with complex culture methods. Meanwhile, various delivery methods have been developed in cell and gene therapy, each presenting distinct advantages and limitations. However, extremely low cell retention rates and survival rates, as well as uneven spatial distribution, significantly constrain therapeutic efficacy. Ongoing research and optimization of advanced biomaterials (e.g., hydrogels, scaffolds) and tissue engineering approaches hold great potential for overcoming these limitations. Therefore, cardiac cell therapy is expected to develop into a comprehensive therapeutic platform that integrates optimized cell sources, intelligent delivery systems, and precisely regulated release mechanisms. Currently, gene therapy is extensively investigated across a wide range of diseases. However, the vector system remains a central challenge. AAV, the primary vector used in gene therapy, exhibits limited targeting efficiency, as only a small fraction successfully reaches the organ. This frequently necessitates the use of higher vector doses, which can directly intensify the cellular immune response and further undermine therapeutic efficacy. Additionally, gene editing technologies such as CRISPR-Cas9 still exhibit off-target effects that may lead to unintended genetic mutations. Moreover, both the overexpression and inadequate regulation of gene expression can lead to adverse outcomes. Therefore, achieving precise and controllable gene expression remains a critical research priority that demands accelerated advancement to transition from conventional gene therapy to precision-based therapeutic strategies. Given the complexity and variability of pathological environments in cardiovascular diseases, the development of multi-gene targeted therapies, potentially integrated with complementary approaches such as engineered cell therapy, pharmacological treatment, or mechanical support, represents a promising avenue for future therapeutic advancements. The current field of cardiac tissue engineering and regenerative medicine is undergoing a paradigm shift, from traditional replacement-based repair toward biomimetic construction and precise regulatory control. This highlights its significant potential while underscoring the major challenges that remain to be addressed. For example, both hydrogel-based materials and 3D bioprinting technologies can effectively replicate the macroscopic physical architecture of the heart. However, they still fall short of the native myocardium in terms of cellular maturity, electrophysiological synchronization, and mechanical contractility. A key underlying cause lies in the absence of a dynamic and coordinated spatiotemporal regulatory mechanism, which increases the risk of arrhythmogenic foci formation and impedes the attainment of synchronous contraction. In this context, self-organizing cardiac organoids demonstrate superior performance. Nevertheless, current cardioid technologies generate structures that more closely resemble immature embryonic tissues. Promoting the maturation of tissue architecture and function, as well as achieving organ-level vascularization, remains an urgent challenge that requires further optimization.

Future research should not only deepen our understanding of the regulatory mechanisms underlying heart development and maturation but also emphasize interdisciplinary collaboration to integrate advanced technologies from developmental biology, genetics, materials science, and other fields. Strengthening the connection between animal models and clinical trials will validate therapeutic efficacy and safety, accelerating the translation of treatments from lab to clinic and improving patient outcomes. These efforts could drive breakthroughs in cardiac regenerative medicine and usher in a new era for heart disease management.

## Figures and Tables

**Figure 1 cells-14-01390-f001:**
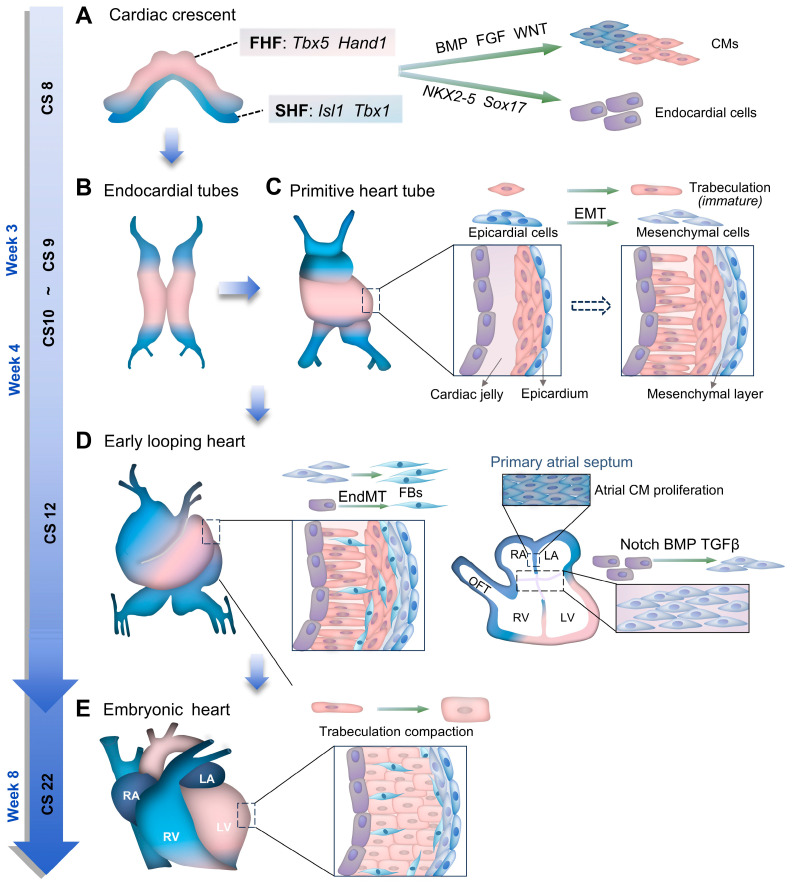
Cartoon illustrating cardiogenesis. (**A**) Precardiac mesoderm cells constitute the cardiac crescent with the first heart field (FHF) and second heart field (SHF) at CS 8, and subsequently differentiated and proliferated into a variety of classes of cells in response to different signals. Specifically, FHF and SHF are roughly distinguished by the expression of transcription factors *Tbx5* and *Hand1*, as well as *Isl1* and *Tbx1*. Cardiomyocytes (CMs) are differentiated under the regulation of signals such as bone morphogenetic protein (BMP), fibroblast growth factor (FGF), and wingless type (WNT) families of growth factors, etc., from adjacent endoderm, and the transcription factors *NKX2.5* and *Sox17* are the initiators of differentiation into endocardial cells. (**B**,**C**) Endocardial tubes form via cardiac crescent remodeling and merge into the primitive heart tube at CS 9. Then it expands, twists, and turns to the right, the cardiac jelly dissipates, and immature trabeculae (long and slender) emerge through myocardial protrusions from the endocardial side. Some epicardial cells contribute to the formation of the epicardium, whereas others transform the mesenchymal layer via the process of epithelial–mesenchymal transition (EMT). (**D**) With further development, mesenchymal layer cells invade the myocardium via endothelial-to-mesenchymal transition (EndMT) to fibroblasts (FBs), and the early looping heart is divided into four chambers by atrial septation, atrioventricular valve, and ventricular septation formation at CS 10–22. The atrioventricular and outflow cushions are expanded through extracellular matrix (ECM) deposition between the endocardial layer and myocardium. Endocardial cells transform into mesenchymal cells through Notch, BMP, and TGFβ signaling pathways, subsequently overlaying the cushions. The primary atrial septum is developed by atrial CM proliferation at CS 12. (**E**) The trabecula becomes shorter and thicker, called trabeculation compaction, and completes at CS 22, developing into the embryonic heart.

**Figure 2 cells-14-01390-f002:**
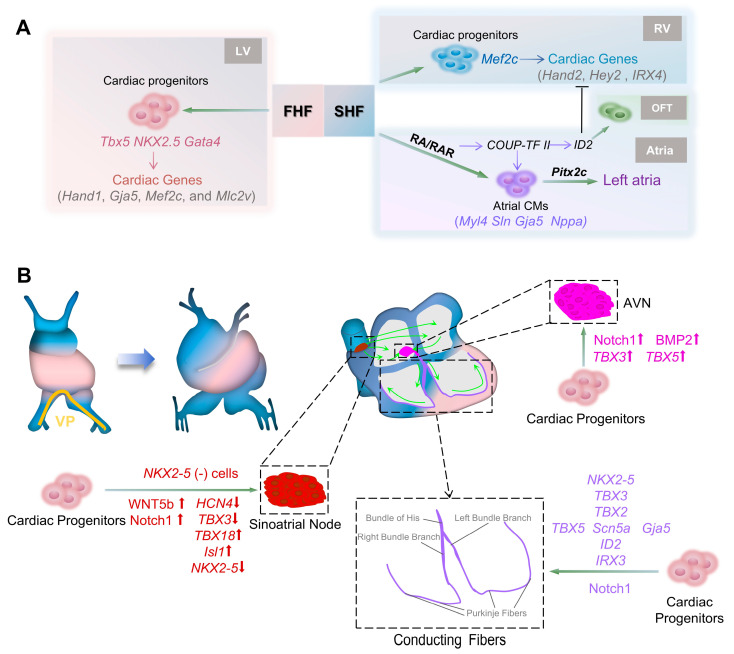
Differentiation and signaling regulation of cardiomyocytes and the conduction system. (**A**) Cardiac progenitors derive from the first heart field (FHF) and second heart field (SHF), and express distinct feature genes. FHF cells form the primitive left ventricle (LV), with left ventricular cardiomyocytes (vCMs) expressing genes such as *Hand1*, *Gja5*, *Mef2c*, and *Mlc2v*. SHF gives rise to the primitive right ventricle (RV), atrium, and outflow tract (OFT), with right vCMs expressing genes such as *Hand2*, *Hey2*, and *IRX4*. Retinoic acid (RA) and its receptor RAR drive atrial cardiomyocyte (aCM) differentiation via *COUP TF II*, leading to the expression of Myosin Light Chain 4 (*Myl4*), Sarcolipin (*Sln*), Gap Junction Protein Alpha 5 (*Gja5*), and Natriuretic Peptide A (*Nppa*). Left atrial development is regulated by *Pitx2c*. OFT development depends on DNA binding 2 (*ID2*), regulated by COUP TF II, and antagonizes right vCM differentiation. (**B**) Conduction system formation and its regulators. The conduction system develops at CS 10 with the primitive heart tube, showing pulsatile contraction driven by the venous pole (VP, yellow). Cardiac progenitors form the sinoatrial node, atrioventricular node (AVN), and conducting fibers in the atrium, atrial septum, and ventricle under different signals/genes. Sinoatrial node cells exhibit an *NKX2-5*-negative phenotype and are regulated by WNT family member 5b (WNT5b) and Notch signaling, *HCN4*, *TBX3*, *TBX18*, and *Isl1* genes. Electrical impulses travel from the sinoatrial node to the atria (green arrows) and reach the AVN via internodal pathways. Notch1 and bone morphogenetic protein 2 (BMP2) signaling and *TBX3*, *TBX5* expression promote cardiac progenitor differentiation into AVN cells. Impulses then pass through the bundle of His, left/right bundle branches, and Purkinje fibers (green arrows), causing ventricular contraction. Conducting fibers differentiate and function under regulation by *NKX2-5*, the *TBX* family, *ID2*, *IRX3*, and Notch signaling.

**Figure 3 cells-14-01390-f003:**
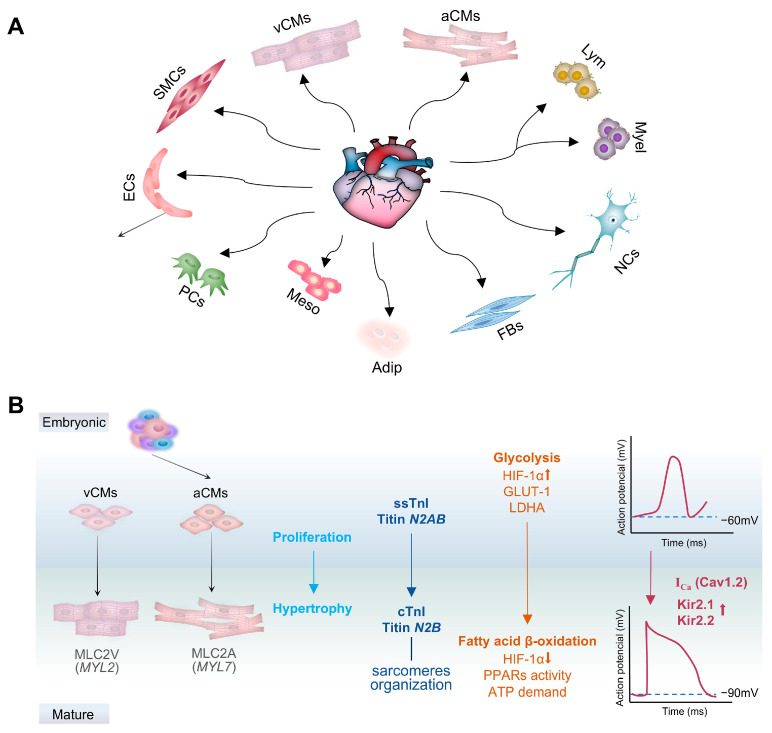
The major cellular component of the mature heart and changes during cardiomyocyte maturation. (**A**) 11 major cell types in adult heart tissue: atrial cardiomyocytes (aCMs), left ventricular cardiomyocytes (vCMs), fibroblasts (FBs), endothelial cells (ECs), smooth muscle cells (SMCs), pericytes (PCs), immune (myeloid, Myel; lymphoid, Lym) cells, adipocytes (Adip), mesothelial cells (Meso), and neuronal cells (NCs). (**B**) Maturation and functional changes of cardiomyocytes. Embryonic cardiomyocytes are immature, characterized by cell proliferation, incomplete sarcomeres, anaerobic glycolysis, and spontaneous contraction. After human birth, cardiomyocytes largely lose their proliferative capacity and undergo physiological hypertrophy; Mature vCMs express myosin heavy chain β (βMHC, encoded by *MYH7*) for myofibril organization, and ventricular myosin light chain 2 (MLC2V, *MYL2*) as a marker. Mature aCMs are marked by atrial myosin light chain 2 (MLC2A, *MYL7*). Sarcomere maturation is achieved by transitioning from slow skeletal muscle troponin I (ssTnI) to cardiac troponin I (cTnI), and Titin splicing from *N2AB* to *N2B*. Cardiomyocyte metabolism shifts from glycolysis to oxidative phosphorylation (fatty acid β-oxidation) at human birth, reducing hypoxia-inducible factor-1α (HIF-1α) expression, which decreases hexokinase, glucose transporter-1 (GLUT-1), and lactate dehydrogenase (LDHA), and to meet the energy supply via peroxisome proliferators-activated receptor (PPAR) family activation. Electrophysiological maturity relies on pacemaker cells and the conduction system, showing Na^+^ influx and membrane depolarization, triggering action potentials and opening L-type calcium channels (Cav1.2) for stable Ca^2+^ currents. Resting membrane potential is maintained by *KCNJ2*/*KCNJ12*-encoded Kir2.1/Kir2.2, reaching −90 mV in mature cardiomyocytes compared to −60 mV in immature cells.

**Figure 4 cells-14-01390-f004:**
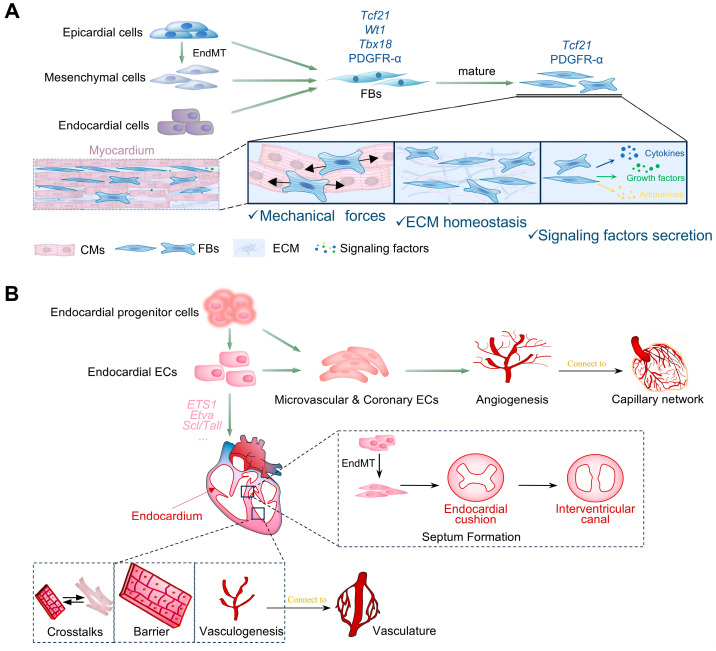
Differentiation and function of cardiac fibroblasts and endothelial cells. (**A**) The origin and roles of cardiac fibroblasts. Embryonic cardiac fibroblasts (FBs) derive from endocardial cells, mesenchymal cells undergoing endothelial-to-mesenchymal transition (EndMT), and epicardial cells. Transcription factor 21 (*Tcf21*), Wilms’ tumor gene 1 (*WT1*), T-box transcription factor 18 (*Tbx18*), and platelet-derived growth factor receptor alpha (PDGFR-α) serve as biomarkers. Mature FBs express only *Tcf21* and PDGFR-α, and play essential roles in the myocardium. FBs provide mechanical forces to surrounding cardiomyocytes (CMs), produce and remodel collagen networks to maintain ECM homeostasis via cytokines, growth factors, and adipokines, ensuring conductivity and rhythmicity. (**B**) The origin, classification, and function of cardiac endothelial cells. Cardiac endothelial cells (ECs) include endocardial ECs, microvascular ECs, and coronary ECs, all derived from endothelial progenitor cells. Endocardial cells can also differentiate into the other two types. *ETS1*, *Etv2,* and *Scl/Tal1* promote endocardial development. ECs cover the entire endocardium to provide a physical barrier, connect with the vasculature through vasculogenesis, and interact with CMs via secreting or transmitting signaling. Microvascular and coronary ECs form capillary networks in the myocardium via angiogenesis, acting as barriers for vascular permeability and tension while supplying oxygen and nutrients to cardiomyocytes.

**Figure 5 cells-14-01390-f005:**
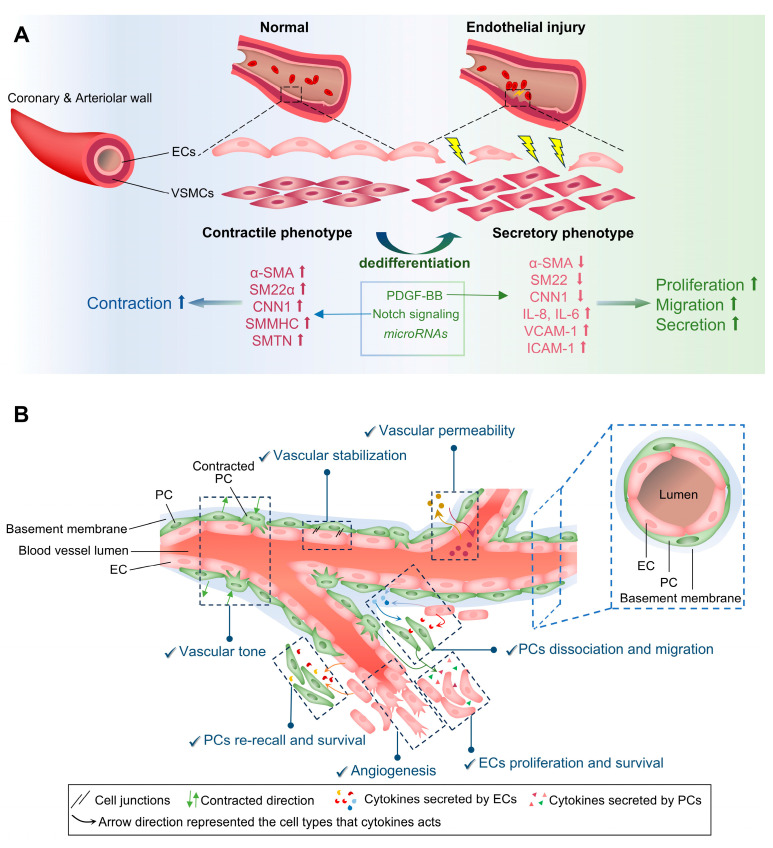
Main distribution and function of cardiac vascular smooth muscle cells and pericytes. (**A**) The distribution and phenotypic transformation of vascular smooth muscle cells. Under physiological conditions, vascular smooth muscle cells (VSMCs) maintain vascular contractility, tone, and integrity through the contractile phenotype, characterized by stretch morphology and markers such as alpha-smooth muscle actin (α-SMA), smooth muscle 22 alpha (SM22α), calponin 1 (CNN1), smooth muscle myosin heavy chain (SMMHC), and smoothelin (SMTN). Notch signaling regulates VSMC stability. In contrast, endothelial injury or platelet-derived growth factor (PDGF) stimulation can drive VSMCs dedifferentiation to a secretory phenotype, reducing contractile markers (α-SMA, SM22α, CNN1) and increasing inflammatory cytokines (Interleukin-8; IL-8, and Interleukin-6; IL-6), vascular cellular adhesion molecule-1 (VCAM-1), and intercellular adhesion molecule 1 (ICAM-1), which enhance cell proliferation, migration, and secretion. (**B**) The distribution and function of pericytes. Pericytes (PCs) are located in the basement membrane of arterioles, venules, and capillaries, where they regulate vascular tone, permeability, and angiogenesis through physical contacts and autocrine/paracrine signaling. This crosstalk between PCs and endothelial cells (ECs) maintains vascular homeostasis. ECs secrete cytokines to promote dissociation, survival, and migration of PCs. In turn, PC-derived cytokines stimulate EC proliferation, survival, and angiogenic sprouting, while endothelium-derived cytokines re-recall PCs for stabilizing new vessels.

**Figure 6 cells-14-01390-f006:**
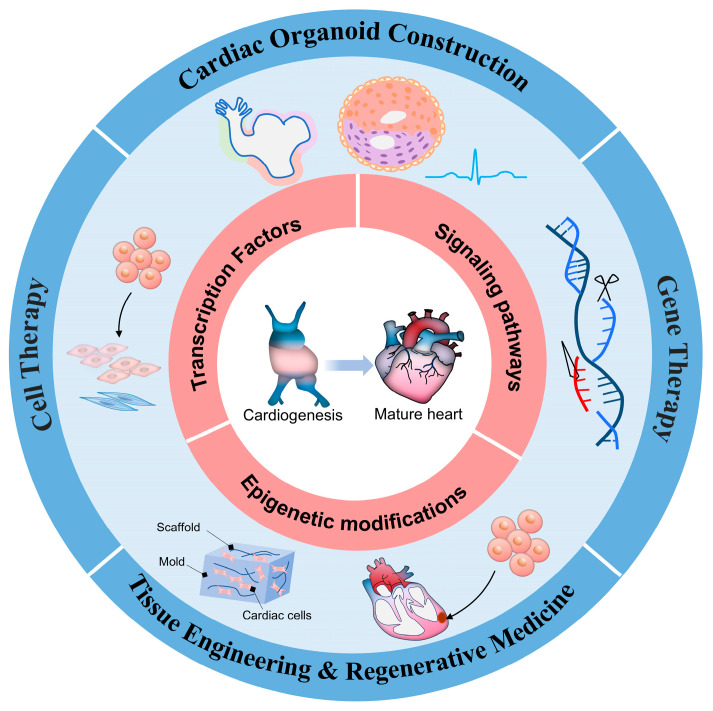
Breakthrough therapeutic advancements through the regulation of cardiogenic factors in cardiogenesis and the mature heart.

## Data Availability

Not applicable.

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
