# Peer review of "Cardiac Development, Cellular Composition and Function: From Regulatory Mechanisms to Applications"

_cells, 2025, doi:10.3390/cells14171390_

Round 1
Reviewer 1 Report
Comments and Suggestions for Authors
This is an interesting and timely article. Most sections are presented in depth, but therapeutic approaches are presented more broadly. The manuscript would benefit by a more focused approach addressing theses three issues.
- Most of the material presented in Sections 1 and 2 about cardiac development recapitulates the review article by Buijtendijk et al. 2020 (ref 1) and could be substantially shortened.
- Section 4 should be expanded since the topiics listed are active areas of research and the number of publications is rather large.
- The role of various exosomes to treat diseased or injured cardiac tissue should be discussed. Examples of therapeutic tissue engineering should be described in more detail.
Specific Comments
- Lines 334-337. These sentences do not accurately reflect the function of the various fibroblast subtypes.
- Lines 385-387 This statement needs further clarification since mechanical stretch of the vessel of cardiac tissue will act on the VSMCs. “VSMCs exhibit high phenotypic plasticity, primarily characterized by a contractile phenotype under physiological conditions, and rarely experience mechanical stress due to the protective layer of ECs.”
- Lines 423-424. “Cardiac pericytes, present in up to 17.2%,“ Clarify whether the percentage refers to the percentage of vessels or percentage of cardiac cells.
Reviewer 2 Report
Comments and Suggestions for Authors
The authors provide an overview on cardiac development and potential repair mechanisms. The review would benefit for more details and a closer connection between the development and repair part. Several major and minor issues could be improved.
Hippo/YAP/TAZ and WT1 signaling are completely missing in the review.
Line 70ff: First and second heart field is a didactic concept. In fact, there is not a single factor known, which is able to distinguish the cells from the first and second heart field clearly.
Figure 1: Please explain all abbreviations. The epicardium, EndMT are not even mentioned in the chapter.
Line 119: “develop various cardiac anatomic structures” – Please be precise.
Line 124: “interactions with other cell types” -which cell types?
Line 127: ECM not explained.
Line 135: “Coronary smooth muscle cells and adventitial fibroblasts originate from epicardium, while the coronary endothelium arises from endocardial and sinus venous endothelial cells[41].” The statement is incorrect. Please see 10.1073/pnas.1509834113.
Line 217: changesFor example- typo
Line 218: activating. Isl1 – typo
Line 274: Cardiomyocyte metabolic switch and end of proliferation largely depend in the timing on the species, which is not mentioned here.
Line 301: involving the modulation of various regulatory factors – which and how?
Line 330: EMC turnover – typo
Line 331: Fibroblasts do not only have passive isolation function but also actively affect cardiomyocyte potentials via mechano-electrical feedback (in the cited references)
Line 334: transform into myofibroblasts
Line 382: SMCs of cardiac Coronary are mainly… grammar and typo
Figure 5 (and all others): spell out full names in the legends and do not introduce in the legend what is not on the scheme e.g. MMPs (matrix metalloproteinase)
Line 420ff: Pericyte coverage of cardiac ECs is only 1:2 – 1:3.
Line 451ff: Clinical trials showed more mixed results than the authors suggest. A more detailed description should be provided (DOI: 10.1038/s41569-024-01098-8).
Line 504ff: For myocardial infarction repair, enhancement of angiogenesis seems reasonable but without stimulation of cardiomyocyte proliferation and limiting the immune overactivation will not help much. Thus, targeting factors, which combine these properties might be more promising (doi: 10.7150/thno.104329).
Organoids: please state clearly the limitation that no organoid approach until now allows cardiac repair or regeneration in humans.
4.4. Regenerative medicine: heart regeneration might be induced by stimulation of cardiomyocyte proliferation (DOI: 10.1038/s41586-023-06585-5) although in this specific case clinical trials using an antagonist had been stopped already many years ago (DOI: 10.1042/CS20060307).
Line 595: “A deeper understanding of the signaling pathways and gene networks during cardiogenesis clarifies disease onset.” The sentence is unclear.
Comments on the Quality of English LanguageEnglish is overall ok, with some minor improvements needed (see above for specific points)
Round 2
Reviewer 1 Report
Comments and Suggestions for Authors
All comments have been adequately addressed.
Reviewer 2 Report
Comments and Suggestions for Authors
The manuscript is significantly improved showing a much more differentiated view of the complex topic. Clinical studies and limitations as well as more recent developments and outlooks are now included. I congratulate the authors on this now highly informative work.